# Ozone Formation during Photocatalytic Oxidation of Nitric Oxides under UV Irradiation with the Use of Commercial TiO_2_ Photocatalytic Powders

**DOI:** 10.3390/ma15175905

**Published:** 2022-08-26

**Authors:** Hubert Witkowski, Wioletta Jackiewicz-Rek, Janusz Jarosławski, Karol Chilmon, Artur Szkop

**Affiliations:** 1Faculty of Civil Engineering, Warsaw University of Technology, 00-637 Warsaw, Poland; 2Institute of Geophysics, Polish Academy of Sciences, 01-452 Warsaw, Poland

**Keywords:** photocatalysis, TiO_2_, numerical simulation, NO_x_, O_3_, photocatalytic powders

## Abstract

The application of photocatalytic materials has been intensively researched in recent decades. The process of nitric oxide (NO) oxidation during photocatalysis has been observed to result in the formation of nitric dioxide (NO_2_). This is a significant factor of the photocatalysis process, as NO_2_ is more toxic than NO. However, it has been reported that ozone (O_3_) is also formed during the photocatalytic reaction. This study analyzed the formation and oxidationof O_3_ during the photocatalytic oxidation of NO under ultraviolet irradiation using commercial photocatalytic powders: AEROXIDE^®^ TiO_2_ P25 by Evonik, KRONOClean^®^ 7050 by KRONOS^®^, and KRONOClean^®^ 7000 by KRONOS^®^. An NO concentration of 100 ppb was assumed in laboratory tests based on the average nitric oxide concentrations recorded by the monitoring station in Warsaw. A mix flow-type reactor was applied in the study, and the appropriateness of its application was verified using a numerical model. The developed model assumed an empty reactor without a photocatalytic material, as well as a reactor with a photocatalytic material at its bottom to verify the gas flow in the chamber. The analysis of the air purification performance of photocatalytic powders indicated a significant reduction of NO and NO_x_ and typical NO_2_ formation. However, no significant formation of O_3_ was observed. This observation was verified by the oxidation of pure ozone in the process of photocatalysis. The results indicated the oxidation of ozone concentration during the photocatalytic reaction, but self-decomposition of a significant amount of the gas.

## 1. Introduction

Construction and building photocatalytic materials have been widely applied to improve air quality in recent decades. The use of photocatalytic materials in asphalt [1,2] and concrete pavements [3,4,5], façade elements [6,7,8,9], and engineering structures such as tunnels [10,11,12] has been described in the literature. Photocatalytic materials have been also widely applied in other processes such as a synthesis of number of novel substances [13]. Studies on the mechanism of photocatalysis building materials have focused on the use of semiconductor materials as a photocatalyst for the removal of ambient airborne pollutants such as nitrogen oxides (NO_x_) and volatile organic compounds (VOCs) [14]. The most commonly studied photocatalyst is titanium dioxide (TiO_2_), which has three crystal structures: anatase, rutile, and brookite. The wide application of TiO_2_ can be attributed to its high photocatalytic activity, chemical stability, safety, relatively cheaper cost, and compatibility with traditional construction materials, such as cement, without any change in its original performance [15]. Catalytic removal of NO_x_ can be also achieved with a selective catalytic reduction with application of Mn–based catalysts, such as Mn–Cu [16,17] or vanadium–based catalysts [18]. However, the process requires a reaction temperature of 280–500 °C to remove nitrogen oxides [19].

The photocatalytic reaction of TiO_2_ was first described by Fujishima and Honda [20] in 1972. During this reaction, irradiation of semiconductor (TiO_2_) with light (λ < 415 nm) causes the electrons (e^−^) to shift from the valence band (*vb*) to the conduction band (*cb*), leading to the formation of electron holes h^+^ (Equation (1), Figure 1). The resulting pairs of charges initiate a reduction–oxidation process. Oxygen and water absorbed from the air act as strong oxidants with the potential to decompose a wide range of compounds (Equations (2) and (3)) [21].
(1)$${\mathrm{TiO}}_{2}\overset{\mathrm{hv}}{\to }{\mathrm{TiO}}_{2}+h_{\mathrm{vb}}^{+}+e_{\mathrm{cb}}^{-},$$
(2)$$H_{2}O_{\mathrm{ads}}+h^{+}\to H^{+}+{\mathrm{HO}}^{∙},$$
(3)$${\left(O_{2}\right)}_{\mathrm{ads}}+e^{-}\to O_{2}^{∙-},$$

The effectiveness of air purification by photocatalytic materials is assessed by laboratory methods of analysis of nitric oxide (NO) oxidation under ultraviolet (UV) light. Nitric oxide is usually chosen as an air pollutant, as it forms nonvolatile products in the presence of a photocatalyst. The most common methodology for testing the air purification performance of photocatalytic materials is described in standards ISO 22197-1:2016 [22], UNI 11247:2010 [23], and JIS TR Z0018:2002 [24]. In these methods, the investigated sample is tested in a laminar flow-type photoreactor activated by UV light with constant gas flow parallel to the surface of the photocatalytic sample through a narrow (5.0 mm ± 0.5 mm [22]) gap. According to the above-mentioned standards, the key parameters determining the air purification performance of photocatalytic materials are gas flow and concentration, sample size, and light source.

The study by Hassan et al. [25] on the flow rate of pollutants during NO_x_ oxidation indicated faster flow rates, lower reduction, and lesser contact time for the photocatalytic reaction. In this study, the authors evaluated the efficiency of NO_x_ removal on a hot mixed asphalt sample treated with a TiO_2_ surface coating. In the study by Asadi et al. [26], the change of gas flow from 3 to 9 L/min caused a reduction in removal efficiency from 62% to 19%. The efficiency of pollutant removal by permeable pavement concrete samples with a TiO_2_ layer has also been analyzed. Similar results were also observed by Martinez et al. for a VOC pollution mixture consisting of benzene, toluene, ethylbenzene, and xylene [27]. The research material used in their study was a TiO_2_ coating applied to cement mortar. The reactant conversion ratio was measured at different flow rates, namely 0.1, 0.4 and 1.5 L/min, of which the greatest pollutant conversion was observed for 0.1 L/min and the lowest for 1.5 L/min. Another key parameter is the pollutant concentration. As stated by Husken et al. [28], increasing inlet concentrations result in lower degradation rates, while with lower pollutant concentrations the rate of oxidation is higher. In the cited study [28], a double-layer paving element obtained from different producers was used. Martinez et al. [8] also emphasized that the effectiveness of oxidation decreases with increasing pollutant concentration. In their study [8], the degradation of NO during photocatalytic oxidation was measured using photocatalytic coatings applied to mortars and glass substrates. The impact of physical parameters on the efficiency of NO_x_ removal has been widely analyzed [29] using different test setups [30,31,32].

Oxidation of NO leads to the generation of nitric dioxide (NO_2_), which is 5–25 times more toxic than the parent compound [33]. Therefore, it is essential to perform an assessment of selectivity. Folli and Macphee [34] described a selectivity parameter for nitrate. Selectivity can be expressed as the ratio of degraded NOx (to nitrogen dioxide ratio (Equation (4)).
(4)$$S=\frac{\Delta {\mathrm{NO}}_{x}}{\Delta \mathrm{NO}}$$

However, NO_x_ oxidation can lead to the formation of various by-products other than NO_2_, such as ozone (O_3_) [29]. It has been observed that ozone formation is facilitated by high concentrations of airborne pollutants along with nitrogen oxides in natural conditions. Although the presence of ozone in the stratosphere is beneficial, as it absorbs UV radiation, in indoor and outdoor conditions it constitutes one of the most critical inorganic pollutants [35]. Ozone is a highly reactive gas with many adverse health effects. Ground –level ozone has a negative impact on health. Even short–term exposure reduces respiratory function by causing asthma or chronic obstructive respiratory disease [36]. Cho et al. [31] investigated the decomposition of O_3_ during photocatalytic and catalytic reactions using TiO_2_ (P25) and Pt-loaded TiO_2_. Their results revealed that a significant amount of ozone is formed during the photocatalytic reaction.

The present study investigated the oxidationof NO_x_ in the photocatalysis process and analyzed the formation of O_3_ and abatement of O_3_. The aim of this study was to determine the amount of ozone generated by NO_x_ oxidation during the photocatalytic reaction. Therefore, abatement of O_3_ in photocatalysis process was performed to verify the effect of ozone formation at the NO oxidation. The paper describes a method for evaluating the air purification performance of photocatalytic powders, and presents a numerical simulation of pollutant reduction in a reactor. To assess the maximal scale of ozone formation during nitric oxide oxidation under UV irradiation, pure photocatalytic powders were used.

## 2. Materials and Methods

### 2.1. Photocatalytic Powders

Three commercial well-known photocatalytic powders (anatase-TiO_2_) [37,38,39] were tested: AEROXIDE^®^ TiO_2_ P25 by Evonik (referred to as P25), KRONOClean^®^ 7050 by KRONOS^®^ (referred to as K7050), and KRONOClean^®^ 7000 by KRONOS^®^ (referred to as K7000). Pure silica powder was used as a reference. The main physicochemical properties of the photocatalysts used in the study are presented in Table 1.

A measured amount (8.00 ± 0.01 g) of photocatalytic powder and silica was evenly spread on a Petri dish (diameter 145 mm) Then, the sample was placed at the bottom of the reactor.

### 2.2. Test Setup

A schematic diagram of the test setup is presented in Figure 2. All parameters, including test time set, flow rate, and concentration of specific gases, were controlled by CGM200 software (MCZ Umwelttechnik, Bad Nauheim, Germany). The set pollutant mixture was provided from the zero air gas generator model Zero Gas Supply NGA 19S (MCZ Umwelttechnik, Bad Nauheim, Germany) and NO from the gas cylinder (Air Liquide, Paris, France), while O_3_ was generated with the ozone generator model Referenz Photometer (MCZ Umwelttechnik, Bad Nauheim, Germany). The specific proportions of pollutants were set in the mass flow controller model CGM 2000 (MCZ Umwelttechnik, Bad Nauheim, Germany). The final gas pollutant was provided to the reactor cell. Then, NO_x_ was measured with NO_x_ analyzer Model T400 (TELEDYNE API, San Diego, CA, USA) and O_3_ with ozone analyzer Model T200 (TELEDYNE API San Diego, CA, USA). The results were collected in data logger model DAS DM370 (DAC SYSTEM, Gdansk, Poland).

#### 2.2.1. Simulation of Gas Flow in the Reactor

The testing of photocatalytic activity in the laboratory condition was performed in a laminar flow reactor, as described in the Standard [22] or in a mix flow reactor assuming complex turbulent gas flow [40]. Zouzelka and Rathousky [30] analyzed the photocatalytic activity of a photocatalytic paint in two types of flow reactors—laminar flow and mix flow reactor. Although the achieved reaction rates were almost comparable for both reactors and the same volume per time of air was purified, the ratio of reactor volume to the area of the irradiated photocatalytic coating was different. The significantly greater volume-to-irradiated area ratio in the mixed reactor was compensated by a longer residence time.

In the present study, a mix flow reactor made of borosilicate glass with a quartz glass cover (200 × 200 × 100 mm) was used. A numerical model was developed to verify mix flow. The scope of the model was gas behavior in the reactor in the presence of a photocatalytic material with absorption capacity and a non-photocatalytic material without absorption capacity. Using this model gas behavior was verified in both the empty reactor (with an object lacking photocatalytic properties) and in the presence of a photocatalytic material. The tracer gas used in the inlet was methane, the molecular weight of which is lower than the average weight of air molecules. In the model, the reactor was of a cuboid shape and made of impermeable walls and a single coherent topology object was placed at its bottom. In the study, the object was assumed to be a cylinder with a diameter of 145 mm and a height of 20 mm. In the model, the chamber was initially filled with pure air. The inlet gas flow was set constant throughout the study. Simulation was carried out using the universal open-source OpenFoam environment, which allows performing numerical simulations based on a wide range of algorithms (solvers) in the scope of fluid mechanics and continuum mechanics. The reactingFoam solver was selected for calculations as it allows modeling of the dynamics and chemical processes associated with the mixing of typical atmospheric gases and lightweight organic compounds (N_2_, O_2_, H_2_O, CO_x_, NO_x_, O_3_). 

In the presented model, the gas flow was analyzed in both the empty reactor (with an object lacking photocatalytic properties) and the reactor with photocatalytic material for which a chemical reaction was not taken into consideration. The effectiveness of the photocatalytic reaction was assumed based on the results of preliminary tests—40% reduction of gas concentration and parametrized as a gradient of the tracer gas near the surface of the photocatalytic sample The selected solver, which works on the basis of the SIMPLE thermodynamic algorithm, analyzed changes in gas pressure, including heat transfer (conduction, diffusion and radiation), with simulation stability at high Reynold’s numbers (even >2100). The geometry of the tested objects was mapped, and the mesh nodes of the object’s surface were interpolated. During model development, reactor configuration, object topology, and flow rates were validated. According to the specification, the applied algorithm maintains stability up to a Courant number value of <0.9. The Courant number is defined as:(5)$$C\;=\left(u\;\Delta t\right)/\Delta x$$
where u is the characteristic velocity in the inlet, t is the numerical time step of the model, and x is the model mesh constant.

#### 2.2.2. Physical Parameters of the Test

Literature data indicate that the typical pollutant concentration to be assumed in laboratory tests on nitric oxide reduction under UV irradiation [3,41,42,43] is 1 ppm. According to the annual report of the District Inspectorate for Environmental Protection in Warsaw published in 2017 [44], the average 1-h concentration of NO_x_ measured by the station in the city center was 103.32 ppb (201 µg/m^3^) and the average 8-h concentration of ozone was 59.61 ppb (121 µg/m^3^). Therefore, in this study, the concentration of NO and O_3_ was assumed at 100 ppb. As the investigated photocatalytic material was a pure powder, the gas flow rate was limited to 2 L/min to not blow out the tested material in the reactor. The humidity in the reactor was limited to 18% considering the form of the tested material. The average temperature was 20 °C. The UV light source used was 20 LED light strips with a wavelength of 365 ± 5 nm (MEISSA Warsaw, Poland), with the maximum light intensity of a single strip being 12 W/m. The intensity of the applied UVA light was 0.1 W/m^2^. Irradiation by UVA as well as global irradiance was constantly measured during the study.

## 3. Results

### 3.1. Simulation of the Photocatalysis Process in the Reactor

The results of the test simulation indicated that to maintain the numerical stability of the complex topology of the assumed object, the Courant number should be reduced to 0.45. Therefore, the spatial resolution of the model had to be increased and consequently the time step had to be shortened. The computation time was approximately proportional to the third power of the spatial resolution. To maintain the value of the Courant number below 0.45, a variable time step was adopted to adjust to the maximum velocities at the particular moment of the simulation. The simulation results showed that the gas concentration in the reactor with non-photocatalytic object was homogenous after 10 min of the experiment (Figure 3). In the case of the reactor with photocatalytic sample, there was a visible reduction of gas concentration after 1 min, as the gas reached the surface of the sample (Figure 4).

### 3.2. Effectiveness of NO_x_ Oxidation

The study assumed the average sample exposure time as minimum 30 min and achieving an NO_x_ concentration stability of <0.5. Stability factor is expressed by the standard deviation value of the last 25 records (every 10 s). Before exposure to UVA light, the samples were kept in the dark for approximately 30 min. Preliminary test indicated that strictly dark conditions should be maintained, as the K7000 photocatalyst is activated by visible light. Even small dosages of light could activate the photocatalytic reaction, and the assumed gas concentration (100 ppb) could not be achieved. The experimental results obtained for P25, K7050, and K7000 samples and silica fume are presented in Figure 5, Figure 6, Figure 7 and Figure 8. The results indicated a significant reduction in the concentration of NO by photocatalytic powders (K7050—58.42%, K7000—62.79%, P25—65.71%) and negligible reduction by silica fume. 

Although results of activity of TiO_2_ samples described in literature [45,46] with different anatase-to-rutile ratio in oxidation process indicate a higher oxidation activity of samples with a higher anatase-to-rutile ratios, achieved results in the study of higher oxidation activity of P25 which is a mix of rutile and anatase over pure anatase samples can be explained with synergism between rutile and anatase particles, what was described by Ohno [47]. The reduction of NO concentration occurred instantly after the UVA light source was turned on, as evidenced by the graphs and values of the NO_x_ stability factor. During oxidation, NO_2_ was generated at a concentration of 4.9 ppb for K7050, 2.5 ppb for K7000. and 4.8 ppb for P25, whereas no generation of NO_2_ was observed for silica fume. The amount of O_3_ generation was not significant for any sample, and the concentration of ozone was found to be negligible in NO reduction tests.

### 3.3. Effectiveness of O_3_ Oxidation

Although numerous preliminary tests were conducted, the concentration achieved in the chamber was lower than that set at the inlet. The measured concentration of ozone in dark conditions for 100 ppb at the inlet was on average 65 ppb. The experimental results obtained for P25, K7050, and K7000 samples and silica fume are presented in Figure 9, Figure 10, Figure 11 and Figure 12. The results indicated that the reduction of ozone concentration (K7050—6.45%, K7000—4.85%, P25—0.87%) was much lower than in the case of nitrogen oxides. No evident oxidation of ozone concentration was observed for silica fume. Despite a lower level of reduction, similar to NO_x_, the process of oxidation began immediately after UVA irradiation.

## 4. Discussion

A comparison of NO and NO_x_ oxidation, NO_2_ formation, and O_3_ oxidation during the oxidation process of specific photocatalytic powders is shown in Table 2. The results of selectivity (S) are also presented in the table. As emphasized by Bloch et al. [33], a high selectivity means almost complete oxidation of NO to nitrate, before it is released into the environment, and strong suppression of the release of NO_x_.

Although a significant reduction of NO concentration was achieved during the photocatalytic reaction of photocatalytic powders under UVA irradiation, no significant formation of ozone was observed during the oxidation process. To explain this phenomenon, an analysis of self-reduction of pure ozone was carried out. The results revealed that pure ozone was reduced in the photocatalytic reaction under UVA irradiation. Nevertheless, such a mechanism was not observed during the oxidation of NO, as no significant increase or decrease of O_3_ concentration occurred. The values of ozone concentration were at the detection range. Therefore, an empty reactor was filled with ozone under UVA irradiation for 1 h. The results of partial self–reduction are presented in Figure 13. No significant decline in ozone concentration under UVA irradiation was observed; however, assuming an initial ozone concentration of 100 ppb, there was significant self-reduction in the reactor.

## 5. Conclusions

This study analyzed the formation and oxidation of O_3_ during photocatalytic oxidationof NO under UV irradiation using commercial photocatalytic powders. An NO concentration of 100 ppb was assumed in laboratory tests based on the average nitric oxide concentrations recorded by the monitoring station in Warsaw. A mix flow-type reactor was applied in the study, and the appropriateness of its application was verified using a numerical model. The results of the simulation proved complete gas mixing in the reactor with non-photocatalytic sample and a significant reduction of the sample placed at the bottom of the reactor. The analysis of the air purification performance of photocatalytic powders indicated a significant oxidation of NO and NO_x_, typical of NO_2_ formation. The results indicated a reduction of over 50% of the initial pollutant concentration. However, no formation of O_3_ was observed. This observation was verified by the oxidation of pure ozone in the process of photocatalysis. The results indicated the reduction of ozone concentration during the photocatalytic reaction, but self-decomposition of a significant amount of the gas. Self-reduction was also observed in the empty reactor irradiated with UVA light.

These findings suggest that, at the assumed NO concentrations and gas flow ratio, ozone formation is completely reduced either by the photocatalysis process itself or by the self-decomposition of the gas. Therefore, the application of building photocatalytic materials with a titanium dioxide catalyst is not related with the ozone formation NO_x_ oxidation process.

## Figures and Tables

**Figure 1 materials-15-05905-f001:**
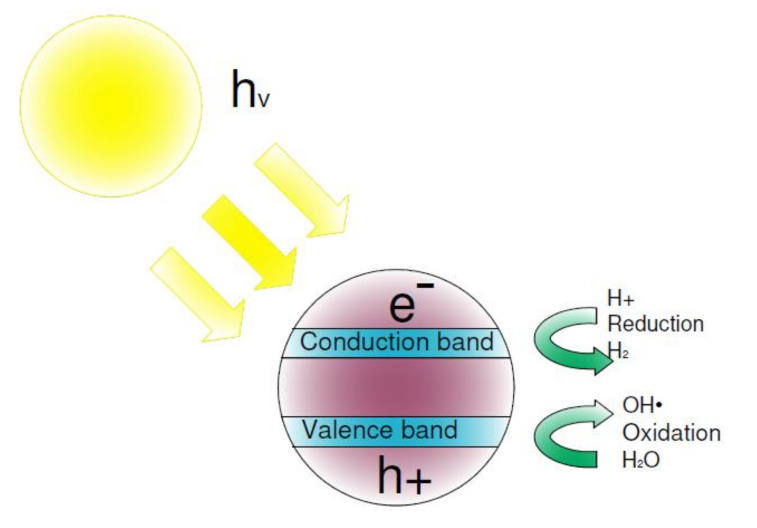
A schematic mechanism of the photocatalytic process.

**Figure 2 materials-15-05905-f002:**
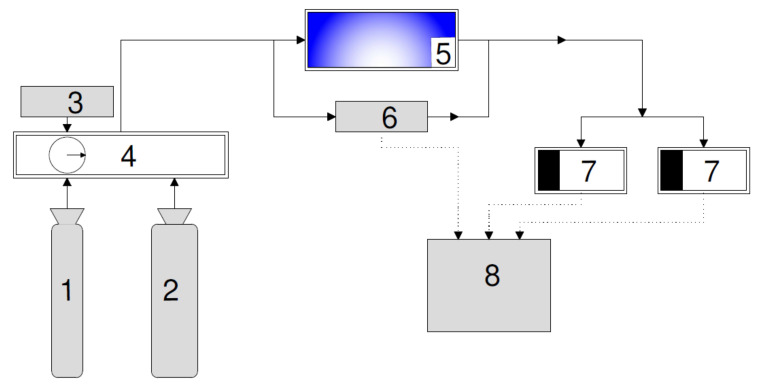
A schematic diagram of the test setup used for NO_x_ and O_3_ reduction: (**1**) NO source, (**2**) zero air generator, (**3**) ozone generator, (**4**) multigas calibrator, (**5**) reactor, (**6**) mass flow controller, (**7**) NO_x_ and O_3_ analyzer, and (**8**) data logger.

**Figure 3 materials-15-05905-f003:**
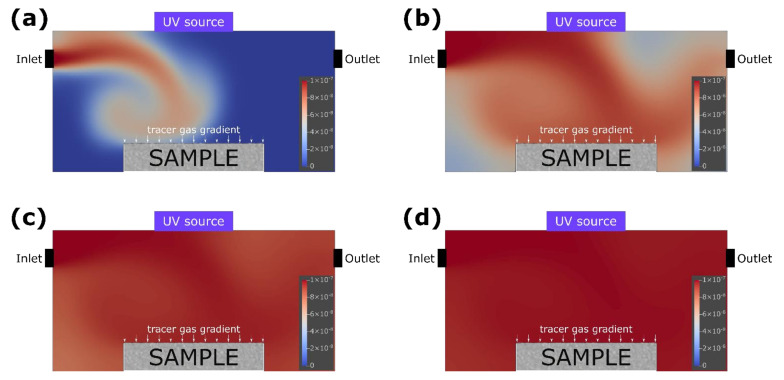
The relative concentration of the tracer gas in the reactor with nonphotocatalytic object, presented on a two-dimensional cross section of the computational domain in the axes of the inlet and outlet at specific time steps: (**a**) 10 s, (**b**) 1 min, (**c**) 3 min, and (**d**) 10 min.

**Figure 4 materials-15-05905-f004:**
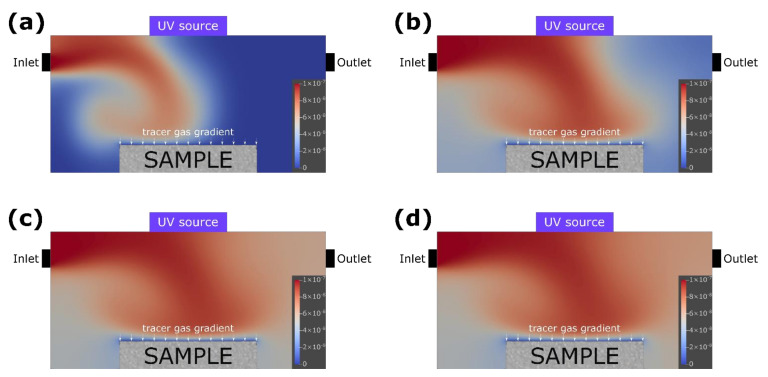
The relative concentration of the tracer gas in the reactor having a photocatalytic sample at the bottom presented on a two-dimensional cross section of the computational domain in the axes of the inlet and outlet at specific time steps: (**a**) 10 s, (**b**) 1 min, (**c**) 3 min, and (**d**) 10 min.

**Figure 5 materials-15-05905-f005:**
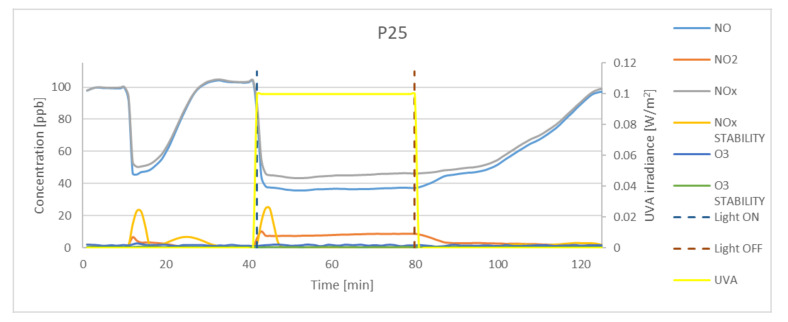
The oxidation of NO in a photocatalytic reaction with concentrations of NO_2_, NO_x_, and O_3_ measured for P25. Data on UVA irradiation are shown in the supplementary chart. Dashed lines indicate the time of light on and off.

**Figure 6 materials-15-05905-f006:**
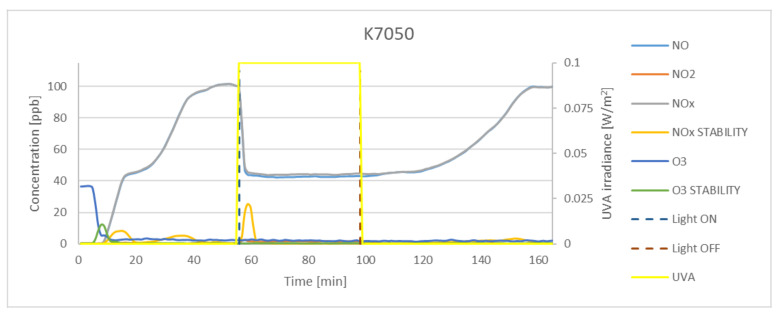
The oxidation of NO in the photocatalytic reaction with concentrations of NO_2_, NO_x_, and O_3_ measured for K7050. Data on UVA irradiation are shown in the supplementary chart. Dashed lines indicate the time of light on and off.

**Figure 7 materials-15-05905-f007:**
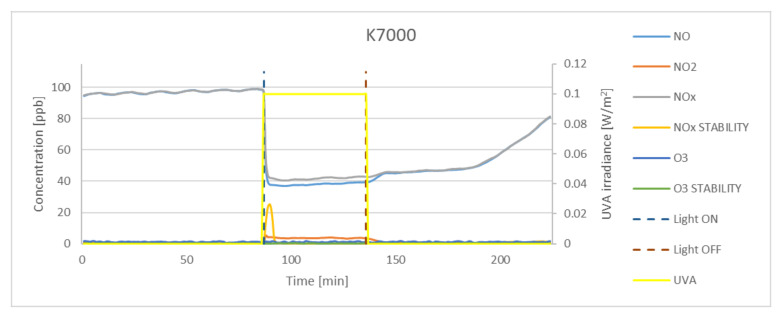
The oxidation of NO in the photocatalytic reaction with concentrations of NO_2_, NO_x_, and O_3_ measured for K7000. Data on UVA irradiation are shown in the supplementary chart. Dashed lines indicate the time of light on and off.

**Figure 8 materials-15-05905-f008:**
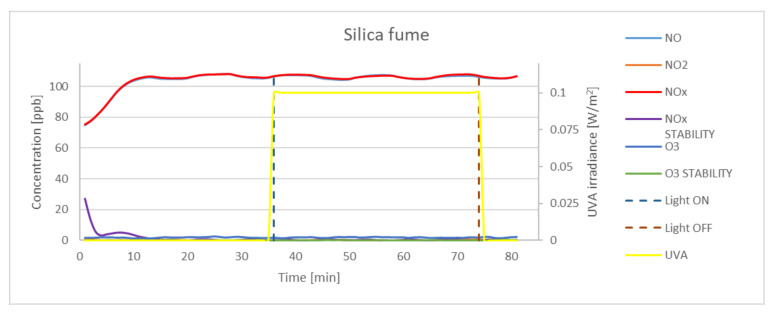
The oxidation of NO in the photocatalytic reaction with concentrations of NO_2_, NO_x_, and O_3_ measured for silica fume. Data on UVA irradiation are shown in the supplementary chart. Dashed lines indicate the time of light on and off.

**Figure 9 materials-15-05905-f009:**
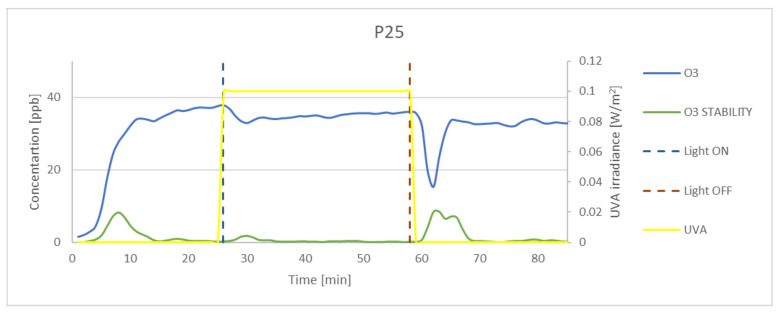
The oxidation of O_3_ in the photocatalytic reaction with concentrations measured for P25. Data on UVA irradiation are shown in the supplementary chart. Dashed lines indicate the time of light on and off.

**Figure 10 materials-15-05905-f010:**
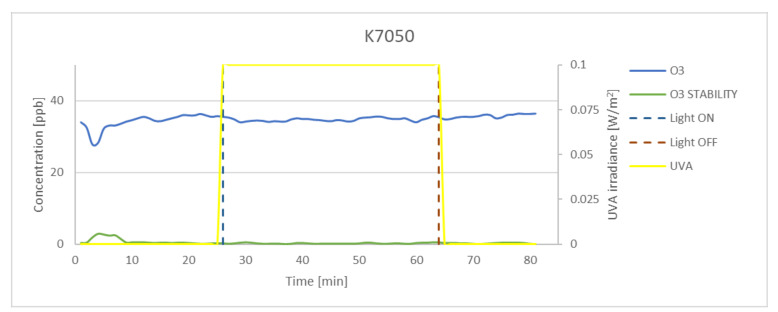
The oxidation of O_3_ in the photocatalytic reaction with concentrations measured for K7050. Data on UVA irradiation are shown in the supplementary chart. Dashed lines indicate the time of light on and off.

**Figure 11 materials-15-05905-f011:**
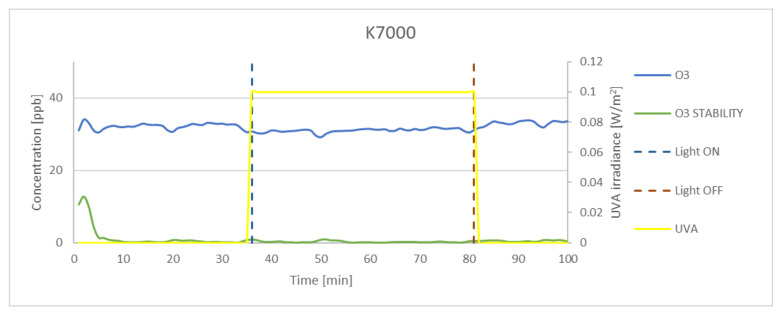
The oxidation of O_3_ in the photocatalytic reaction with concentrations measured for K7000. Data on UVA irradiation are shown in the supplementary chart. Dashed lines indicate the time of light on and off.

**Figure 12 materials-15-05905-f012:**
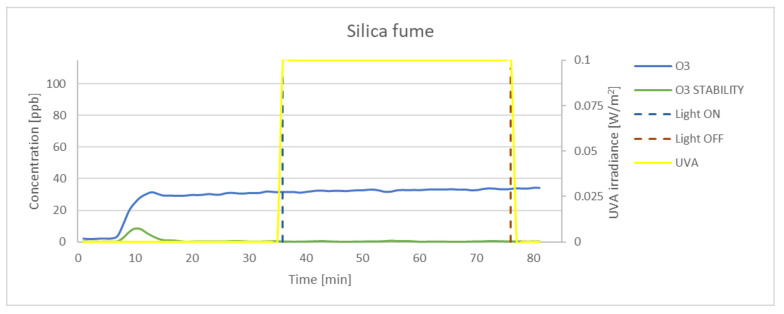
The oxidation of O_3_ in the photocatalytic reaction with concentrations measured for silica fume. Data on UVA irradiation are shown in the supplementary chart. Dashed lines indicate the time of light on and off.

**Figure 13 materials-15-05905-f013:**
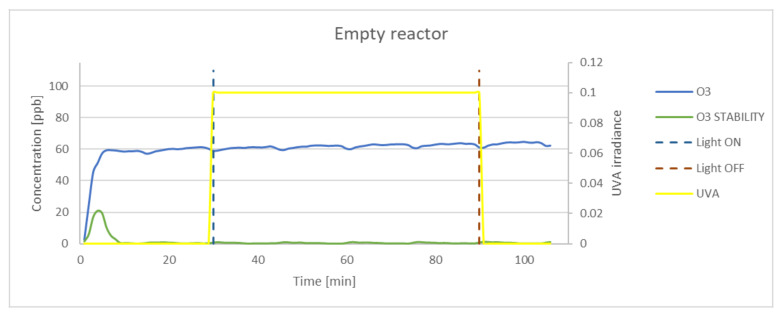
The oxidation of O_3_ in the empty reactor under UVA irradiation. Data on UVA irradiation are shown in the supplementary chart. Dashed lines indicate the time of light on and off.

**Table 1 materials-15-05905-t001:** Characteristics of the commercial TiO_2_ powders and reference silica (information provided by the supplier).

Tested Material	Crystal Modification	Density [g/cm^3^]	Bulk Density [g/L]	Specific Surface Area (BET) [m^2^/g]	Crystallite Size (Anatase) [nm]
P25	Anatase (87%) and rutile (13%) *	4.26	100–180 *	53.8 ± 0.2	33
K7050	Anatase	3.9	300	326 ± 2.6	8
K7000	Anatase	2.9	350	246.8 ± 2.9	10
Silica	Silica	2.2	225	23.86 ± 0.69	-

* Tamped density ex plant.

**Table 2 materials-15-05905-t002:** Results of the oxidation of NO and O_3_ in the photocatalytic reaction under UVA irradiation for the studied powders and silica fume. NO, NO_2max_, and NO_x_ refer to NO oxidation, O_3_ refers to pure ozone oxidation, and S refers to the calculated selectivity of the NO reduction process.

Tested Material	NO			O_3_	S
	NO [%]	NO_2max_ [ppb]	NO_x_ [%]	O_3_ [%]	[%]
P25	65.71	10.40	58.60	12.96	89.18
K7050	58.42	2.4	57.03	6.34	97.62
K7000	62.79	5.30	59.27	14.37	94.39
Silica	3.51	0.8	2.97	2.21	84.62

## Data Availability

The data are contained within the article.

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
