# Peer review of "Ozone Formation during Photocatalytic Oxidation of Nitric Oxides under UV Irradiation with the Use of Commercial TiO2 Photocatalytic Powders"

_materials, 2022, doi:10.3390/ma15175905_

Round 1

Reviewer 1 Report

This article is devoted to the study of the processes of ozone formation during the photocatalytic decomposition of nitrogen oxides on TiO2 catalysts. The article is written in a clear and accessible language. Abstract is written intelligibly and to the fullest. Introduction is rich in material, which allows you to delve into the research issues. the abundance of methods and descriptions have a good effect on the structure of the materials presented in this work. However, I recommend the authors to make the following improvements:

1. Table 1. For catalyst P25, it is desirable to indicate the percentage of anatase and rutile.

2. How would pure rutile work in this reaction?

3. Are there data in the literature on other transition metal oxides used in this process? if yes, then comparison with them is necessary.

4. The article contains a lot of experimental data and their descriptions. However, references to the literature are lacking. It would be very good if in each description of each method there were references to the literature and comparison with known results.

5. It is desirable that the authors cite the work: 10.1007/s13399-022-02587-x.

6. It is desirable that in the conclusions the authors place more emphasis on the practical use of the results obtained.

Author Response

We would like to thank the editor and the reviewers for their effort to review our manuscript. They raised crucial issues and their contribution is very valuable for improvement in the quality of our manuscript. We generally agree with the editor and reviewer’s comments and we have revised our manuscript accordingly. We did our best to improve the manuscript greatly such that it meets the expectations of the reviewers. Below we reply closely to every received comment. We hope that the editor and the reviewers will acknowledge our responses to their comments as satisfactory. We are ready to finish the revised version of the manuscript concerning any further suggestion that the editor or reviewers may have.

Please find our responses to each editor and reviewer’s comment below.

We are looking forward to hearing from you soon.

Yours sincerely,

Hubert Witkowski

Reviewer #1

  1. Table 1. For catalyst P25, it is desirable to indicate the percentage of anatase and rutile.

Table 1. has been updated with the percentage of anatase and rutile for P25 according to the laboratory tests.

Tested Material

Crystal modification

Density [g/cm3]

Bulk density [g/l]

Specific surface area (BET) [m2/g]

Crystallite size (anatase) [nm]

P25

Anatase (87%)    and rutile (13%)*

4.26

100 – 180**

53.8 ± 0.2

33

K7050

Anatase

3.9

300

326 ± 2.6

8

K7000

Anatase

2.9

350

246.8 ± 2.9

10

Silica

Silica

2.2

225

23.86 ± 0.69

-

  1. How would pure rutile work in this reaction?

Findings of Zhang et al. [1] on photocatalytic efficiency of titanium oxides indicated high efficiency of anatase TiO2 comparing to rutile TiO2 in NO photocatalytic oxidation. Although results of activity of TiO2 samples described in literature [3, 4] with different anatase-to-rutile ratio in oxidation process indicate a higher oxidation activity of samples with a higher anatase-to-rutile ratios, achieved results in the study of higher oxidation activity of P25 which is a mix of rutile and anatase over pure anatase samples can be explained with synergism between rutile and anatase particles, what was described by Ohno [5].

  1. Are there data in the literature on other transition metal oxides used in this process? If yes, then comparison with them is necessary.

Catalytic removal of NOx can be also achieved with a selective catalytic reduction with application of Mn–based catalysts, such as Mn–Cu [6, 7] or vanadium –based catalysts [8]. However the process requires reaction temperature of 280 – 500°C to remove nitrogen oxides [9].  

  1. The article contains a lot of experimental data and their descriptions. However, references to the literature are lacking. It would be very good if in each description pf each method there were references to the literature and comparison with known results.

References has been reviewed and uploaded.

  1. It is desirable that the authors cite the work: 10.1007/s13399-022-02587-x.

Citation has been uploaded.

  1. It is desirable that in the conclusions the authors place more emphasis on the practical use of the results obtained

Conclusions has been updated: These findings suggest that, at the assumed NO concentrations and gas flow ratio, ozone formation is completely reduced either by the photocatalysis process itself or by the self-decomposition of the gas. Therefore application of building photocatalytic materials with titanium dioxide catalyst is not related with ozone formation form NOx oxidation process.

References

[1] Zhang, T. Ayusawa, M. Minagawa, K. Kinugawa, H. Yamashita, M. Matsuoka, M. Anpo, Investigations of TiO2 Photocatalysts for the Decomposition of NO in the Flow System, Journal of Catalysus 198, 1 – 8 (2001), http://doi.org/10.1006/jcat.2000.3076,

[2] Z. Ding, G.Q. Lu., P.F. Greenfield, Role of the Crystallite Phase of TiO2 in Heterogeneous Photocatalysis for Phenol Oxidation in Water, The Journal of Physical Chemistry B 2000, 104,19, pp. 4815-4820, https://doi.org/10.1021/jp993819b,

[3] T. van der Meulen, A. Matson, L. Österlund, A comparative study of the photocatalytic oxidation of propane on anatase, rutile, and mixed – phase anatase – rutile TiO2 nanoparticles: Role of surface intermediates, Journal of Catalysis 2007, 251, 1, pp. 131 – 144, https://doi.org/10.1016/j.jcat.2007.07.002

[4] T. Ohno, K. Tokieda, S. Higashida, M. Matsumura, Synergism between rutile and anatase TiO2 particles in photocatalytic oxidation of naphthalene, Applied Catalysis A: General, V. 244, I. 2, 2003, pp. 383 – 391, https://doi.org/10.1016/S0926-860X(02)00610-5.

[5] X. Ren, Z. Ou, B. Wu, Low – Temperature Selective Catalytic Reduction DeNOx and Regeneration of Mn – Cu Catalyst Supported by Activated Coke, Materials 2021, 14, 5958, https://doi.org/10.3390/ma14205958 ,

[6] D. Urbanas, E. Baltrenaite–Gediene, Selective Catalytic Reduction of NO by NH3 over Mn–Cu Oxide Catalysts Supported by Highly Porous Silica Gel Powder: Comparative Investigation of Six Different Preparation Methods, Catalysts 2021, 11,702, https://doi.org/10.3390/catal1106072,  

[7] B. Yu, Q. Liu, H. Yang, Q. Li, H. Lu, L. Yang, F. Liu, Selective Catalytic Removal of High Concentrations of Nox at Low Temperature, Energies 2022,15, 5433, https://doi.org/10.3390/en15155433,

[8] J. Jang, S. Ahn, S. Na, J. Koo, H. Roh, G. Choi Effect of Plasma Burner on NOx Reduction and Catalyst Regeneration in a Marine SCR System, Energies 2022, 15, 4306, https://doi.org/10.3390/en1512306,

Author Response

We would like to thank the editor and the reviewers for their effort to review our manuscript. They raised crucial issues and their contribution is very valuable for improvement in the quality of our manuscript. We generally agree with the editor and reviewer’s comments and we have revised our manuscript accordingly. We did our best to improve the manuscript greatly such that it meets the expectations of the reviewers. Below we reply closely to every received comment. We hope that the editor and the reviewers will acknowledge our responses to their comments as satisfactory. We are ready to finish the revised version of the manuscript concerning any further suggestion that the editor or reviewers may have.

Please find our responses to each editor and reviewer’s comment below.

We are looking forward to hearing from you soon.

Yours sincerely,

Hubert Witkowski 

Reviewer #2

  1. Manuscript writing is weak and has very ambiguities in the explanation of the purpose of the study, used method and obtained results.

We do hope that reviewed manuscript with valuable reviewer’s comments improved the quality of our manuscript and it meets the expectation of reviewer. 

  1. Is the studied process NO decomposition, NO oxidation or NO reduction? Explanation in the manuscript text related to NO oxidation or NO reduction while in the manuscript title stated as NO decomposition. Conducted process should be described in detail and revised in the manuscript.

We thank very much to the reviewer for this remark. The manuscript has been updated.

  1. It is not determined what reactions have been considered in the simulation as well as their kinetics.

In the presented model, the gas flow was analyzed in both the empty reactor (with an object lacking photocatalytic properties) and the reactor with photocatalytic material for which a chemical reaction was not taken into consideration. The effectiveness of the photocatalytic reaction was assumed based on the results of preliminary tests—40% reduction of gas concentration and parametrized as a gradient of the tracer gas near the surface of the photocatalytic sample.

  1. Figures 4 – 11 are reported as experimental results, why are the simulation results not presented?

The main goal of the presented simulation was a verification of gas flow in the reactor, as the reactor type was different to the one described in the Standards. In the simulation rate of the pollutant concentration reduction was an average of the results achieved in the study, therefore comparison with the simulation has not been presented.

  1. The explanation provided for the equation 4 in the text does not match the mathematical form of the equation.

We thank very much to the reviewer for this remark. The text has been updated.

  1. The aim of the performing ozone oxidation experiments should be clearly stated.

The present study investigated the oxidation of NOx in the photocatalysis process and analyzed the formation of O3 and abatement of O3. The aim of this study was to determine the amount of ozone generated by NOx oxidation during the photocatalytic reaction. Therefore abatement of O3 in photocatalysis process was performed to verify effect of ozone formation at the NO oxidation. The paper describes a method for evaluating the air purification performance of photocatalytic powders, and presents a numerical simulation of pollutant reduction in a reactor. To assess the maximal scale of ozone formation during nitric oxide oxidation under UV irradiation, pure photocatalytic powders were used.   

  1. Ozone generator or supplier was not shown in the Schematic diagram of the test setup.

We thank very much to the reviewer for this remark. The figure has been updated.

8. The quality of the figures 2 and 3 is low.

We have updated figures with higher quality.

Reviewer 3 Report

In current work (Manuscript ID: materials-1847223), the authors carried out detail investigation on the formation and reduction of O3 during photocatalytic decomposition of NO under ultraviolet irradiation using commercial TiO2 photocatalyst powders. The analysis of the air purification performance of photocatalyst powder indicated a significant reduction of NO and NOx and typical NO2 formation. However, no formation of O3 was observed. This observation was verified by the reduction of pure ozone in the process of photocatalysis. The results indicated the reduction of ozone concentration during the photocatalytic reaction, but self-decomposition of a significant amount of the gas. This work is well-organized and contains acceptable level of new results. I will consider its publication in materials after minor revision as noted below.

1. Abstract should be more concise and specific.

2. In introduction part, main problems and their possible solutions should be addressed.

3. Authors are suggested to provide a schematic mechanism which depicts charge carriers’ generation, separation, and surface reactions.

4. Some typo-errors and grammatical mistakes exists throughout the manuscript. The authors should carefully revise the manuscript and do corrections accordingly.

5. The references should be consistent throughout. Especially check titles of all references.

Author Response

We would like to thank the editor and the reviewers for their effort to review our manuscript. They raised crucial issues and their contribution is very valuable for improvement in the quality of our manuscript. We generally agree with the editor and reviewer’s comments and we have revised our manuscript accordingly. We did our best to improve the manuscript greatly such that it meets the expectations of the reviewers. Below we reply closely to every received comment. We hope that the editor and the reviewers will acknowledge our responses to their comments as satisfactory. We are ready to finish the revised version of the manuscript concerning any further suggestion that the editor or reviewers may have.

Please find our responses to each editor and reviewer’s comment below.

We are looking forward to hearing from you soon.

Yours sincerely,

Hubert Witkowski

Reviewer #3

  1. Abstract should be more concise and specific.

Abstract has been updated: The application of photocatalytic materials has been intensively researched in recent decades. The process of nitric oxide (NO) oxidation during photocatalysis has been observed to result in the formation of nitric dioxide (NO2). This is a significant factor of the photocatalysis process as NO2 is more toxic than NO. However, it has been reported that ozone (O3) is also formed during the photocatalytic reaction. This study analyzed the formation and oxidation of O3 during the photocatalytic  oxidationof NO under ultraviolet irradiation using commercial photocatalytic powders: AEROXIDE® TiO2 P25 by Evonik, KRONOClean® 7050 by KRONOS®, and KRONOClean® 7000 by KRONOS®. An NO concentration of 100 ppb was assumed in laboratory tests based on the average nitric oxide concentrations recorded by the monitoring station in Warsaw. A mix flow-type reactor was applied in the study, and the appropriateness of its application was verified using a numerical model. The developed model assumed an empty reactor without a photocatalytic material and a reactor with a photocatalytic material at its bottom to verify the gas flow in the chamber. The analysis of the air purification performance of photocatalytic powders indicated a significant oxidation of NO and NOx and typical NO2 formation. However, no significant formation of O3 was observed. This observation was verified by the  oxidation of pure ozone in the process of photocatalysis. The results indicated the  oxidation of ozone concentration during the photocatalytic reaction, but self-oxidation of a significant amount of the gas.

  1. In introduction part, main problems and their possible solutions should be addressed.

Introduction part has been updated.

  1. Authors are suggested to provide a schematic mechanism which depicts charge carriers’ generation, separation, and surface reactions.

A figure 1 has been provided.

  1. Some typo-errors and grammatical mistakes exists throughout the manuscript. The authors should carefully revise the manuscript and do corrections accordingly.

Manuscript has been revised and corrections has been done accordingly.

  1. The references should be consistent throughout. Especially check titles of all references.

References has been revised and corrections has been done accordingly.